



# Modulation of surface sensible heating over the Tibetan Plateau on the interannual variability of East Asian dust cycle

Xiaoning Xie[1,2], Anmin Duan[3], Zhengguo Shi[1], Xinzhou Li[1], Hui Sun[1], Xiaodong Liu[1,4], Xugeng Cheng[5], Tianliang Zhao[5], Huizheng Che[6], and Yangang Liu[7]

[1]SKLLQG, Institute of Earth Environment, Chinese Academy of Sciences, Xi'an 710061, China
[2]CAS Center for Excellence in Quaternary Science and Global Change, Xi'an 710061, China
[3]LASG, Institute of Atmospheric Physics, Chinese Academy of Sciences, Beijing 100029, China
[4]University of Chinese Academy of Sciences, Beijing 100049, China
[5]Key Laboratory for Aerosol-Cloud-Precipitation of China Meteorological Administration, Nanjing University of Science Information &Technology, Nanjing 210044, China
[6]Key Laboratory for Atmospheric Chemistry, Institute of Atmospheric Composition, Chinese Academy of Meteorological Sciences, CMA, Beijing 100081, China
[7]Environmental and Climate Sciences Department, Brookhaven National Laboratory, Upton, NY 11973-5000, USA

*Correspondence to:* Xiaoning Xie (xnxie@ieecas.cn)

**Abstract.** Previous observational evidence and numerical simulations have revealed that the surface sensible heating in MAM (March-April-May) over the Tibetan Plateau (TPSH) can affect the Asian regional hydrological cycle, surface energy balance, and climate through altering atmospheric heat source of the Tibetan Plateau (TP). This study aims to investigate the impacts of MAM TPSH on the interannual variability of East Asian dust cycle by use of CAM4-BAM (version 4 of the Community

Atmosphere Model coupled to a bulk aerosol model), the MERRA-2 (version 2 of the Modern Era Retrospective-Analysis for Research and Applications) surface dust concentration, and TPSH measurements. Our simulations show that the surface dust concentrations over the East Asian dust source region (EA) and over the northwestern Pacific (NP) in MAM are significantly positively correlated with TPSH, with regionally averaged correlation coefficients of 0.49 for EA and 0.44 for NP. Similar positive correlations are also shown to exist between the MAM TPSH measurements averaged over the 73 observation sites

and the surface dust concentration from MERRA-2. Simulation-based comparisons between strongest and weakest TPSH years reveal that, the MAM surface dust concentration in the strongest TPSH years increases with relative differences of 13.1% over EA and 36.9% over NP. These corresponding differences are found in MERRA-2 with 22.9% and 13.3% over EA and NP, respectively. Further simulated results show that the processes of whole dust cycles (e.g., dust loading, emission, and transport, as well as dust depositions) are also significantly enhanced during the strongest TPSH yeas over EA and NP. Through enhancing

the TP heat source, stronger TPSH in MAM generates an anticyclonic anomaly in middle and upper troposphere over TP and over the downstream Pacific region, respectively. These atmospheric circulation anomalies induced by the increased TPSH result in increasing the westerly winds over both EA and NP, which in turn increases dust emissions over the dust source, and dust transports over these two regions, as well as the regional dust cycles. These results suggest that addressing the East Asian dust changes in the future climates require considering not only increasing greenhouse gas emissions but also the variations of

the TP's heat source under global warming.



# 1 Introduction

East Asia is a major source of global dust aerosols originating from the Taklamakan desert and the Gobi desert. Over this region, the estimated several hundred Tg per year of dusts are emitted directly into the air and partly transported to downstream land and ocean regions through westerly winds, e.g., eastern China and northern Pacific (Duce et al., 1980; Zhang et al., 1997; Sun et al., 2001; Gong et al., 2006; Zhao et al., 2006). Due to high atmospheric dust concentrations (Zhang et al., 2012) and dust optical depth (AOD) over East Asia (Che et al., 2015), the interactions between dust aerosols and climate changes are of fundamental importance from observational and numerical studies (e.g., Liu et al., 2004; Gong et al., 2006; Shao et al., 2011; Huang et al., 2014; Mahowald et al., 2014).

Dust aerosols can affect the solar and infrared radiation budgets at the top of atmosphere (TOA) and at the surface, which in turn influences the regional climate through the dust direct (DRF) and dust-in-snow radiative forcings. Previous studies have revealed an importance of dust DRF on global energy balance in general circulation models (GCMs) in recent decades (Tegen and Lacis, 1996; Miller and Tegen, 1998; Yue et al., 2009; Zhang et al., 2010; Mahowald et al., 2014). As summarized by Kok et al. (2017), it shows that the global average dust DRF was almost $-0.4$ W m$^{-2}$ in the current GCMs of Huneeus et al. (2011). Furthermore, they also claimed that the global average dust DRF was significantly underestimated as a result of less coarser dust aerosols in these current GCMs, which shows a much larger dust DRF of almost $-0.2$ W m$^{-2}$ based on a new size distribution of dust aerosols derived from Kok (2011) (Kok et al., 2017). Due to larger dust loading (Zhang et al., 2012) and dust AOD over East Asia (Che et al., 2015), the average dust DRF over this region is much larger than that for the global mean (Zhang et al., 2009; Sun et al., 2012; Han et al., 2012; Xin et al., 2016), which could further influence the East Asian climate significantly (Lau et al., 2006; Sun et al., 2012, 2017; Gu et al., 2016; Tang et al., 2018). A mechanism of Elevated Heat Pump (EHP) from absorbing aerosols, affects the monsoon atmospheric circulation and the summer monsoon precipitation over the South and East Asia in late spring and early summer (Lau et al., 2006). Based on sensitivity tests of the WRF-Chem model with a detailed dust emission scheme, Tang et al. (2018) also showed the existence of EHP mechanism over the northwestern China, which directly causes regional anomalous circulations. Zhang et al. (2009) have shown that the total radiative forcings at the TOA and at the surface are significantly decreased by the dust DRF, resulting in a surface cooling over East Asia (up to $-1°$C) and increasing the regional local atmospheric stability. The dust DRF changes the summer monsoon precipitation and atmospheric circulations over East Asia through altering atmospheric thermal structures (Sun et al., 2012, 2017; Tang et al., 2018). Furthermore, a comparison between North Africa and South/East Asia, shows that change in the monsoon precipitation induced by the dust DRF is absolutely distinct over these monsoon regions, relying on the relative location of dust aerosols to the precipitation band. Additionally, the surface albedo perturbation due to dust and black carbon in snow causes a strong and positive radiative forcing at the surface from 5 W m$^{-2}$ to 25 W m$^{-2}$ in MAM over the Tibetan Plateau region (TP) (Flanner et al., 2009; Qian et al., 2011, 2015; He et al., 2018; Xie et al., 2018b). These absorbing aerosols deposited on snow warm the TP's surface, and alter the East and South Asia summer precipitation and monsoon atmospheric circulation (Qian et al., 2011; Shi et al., 2019), and the arid climate over the northwestern China through enhancing the TP's elevated heat source (Xie et al., 2018b).





Changes in dust cycle-related processes, including dust emission, dust transport, and dust depositions, are affected by meteorological and climatic parameters (Littmann, 1991; Qian et al., 2004; Liu et al., 2004; Gong et al., 2006; Zhao et al., 2006; Yumimoto and Takemura, 2015; Lou et al., 2016), as well as the dust radiative forcing including dust DRF (Miller et al., 2004; Heinold et al., 2007; Xie et al., 2018a; Cheng et al., 2019) and dust-in-snow radiative forcing (Xie et al., 2018b). According to

the meteorological data, large quantities of the Asian dust storms are generated from high wind speeds associated with cyclonic activities and cold surges (Littmann et al., 1991; Sun et al., 2001; Liu et al., 2004). Littmann (1991) examined relationships of the Asian dust storm frequency to meteorological parameters, and found positive correlations with wind speed and negative correlation with surface precipitation. Qian et al. (2004) have shown that, over the northern China, the regional dust storms show a negative correlation with the prior winter temperature. Furthermore, the spring dust storm frequency is strongly neg-

atively positive with the antecedent annual and seasonal soil moisture and surface precipitation, shown in Liu et al. (2004). Interannual variability in the intensity of the East Asian winter monsoon could not directly affect the dust emissions but the dust transport in observations and models (Gong et al., 2006; Zhao et al., 2006; Yumimoto and Takemura, 2015; Lou et al., 2016). Additionally, the dust DRF reduces the dust emissions through interactions with the planetary boundary layer (PBL) over Saharan and East Asian dust sources at the current climate condition (Miller et al., 2004; Heinold et al., 2007; Xie et

al., 2018a), whereby the net negative DRF at the surface decreases surface sensible heat fluxes into the atmosphere, weakens turbulent mixing within the PBL, and in turn reduces surface wind speeds. The dust DRF-induced reduction in dust emissions is much more significant with the mechanism of PBL during the Last Glacial Maximum period, in comparison with the current climate condition (Cheng et al., 2019). More recently, based on sensitivity of GCM simulations to dust radiative forcings, Xie et al. (2018b) revealed that the dust-in-snow radiative forcing over TP significantly increases the eastern Asian dust emissions

and the regional dust cycle through enhancing westerly winds over the northwestern China.

As we known, the TP region is the largest and highest plateau in the world with the average elevation of around 4.5 km, and is often called as the "third pole" of the Earth in view of its importance in shaping the Earth's climate (Yao et al., 2012). In MAM (March-April-May), the air column over TP shows a significant change from a heat sink to a heat source mainly due to the rapidly enhanced surface sensible heating over TP (labeled by TPSH). The TPSH in MAM reaches its annual maximum flux,

which could dominate the sum of latent and sensible heating fluxes, as well as long-wave and short-wave radiative forcings over the TP (Duan et al., 2017). Therefore, the TPSH can basically determine the overall thermal forcing over TP in MAM. Previous observational and modeling studies haven shown that the MAM TPSH can affect the South and East Asian hydrological cycle (Wu and Zhang, 1998; Duan et al., 2011; Wu et al., 2012; An et al., 2015) and the regional large-scale circulation in middle and upper layers (Ye and Wu, 1998; Liu and Dong, 2013; An et al., 2015; Duan et al., 2017) through altering atmospheric heat

source of the TP. It is likely that the changes in the large-scale circulation induced by the MAM TPSH then influence the East Asian dust cycle by altering dust emissions and dust transports. However, the detailed relationships between TPSH and the East Asian dust cycle remain elusive.

Hence, we aim this study to explore the relationships between the MAM TPSH and the regional dust processes, by combining CAM-BAM simulations, reanalysis, and measurements. The remainder of this paper is organized as follows: Sect. 2 first

describe observational and reanalysis datasets, the updated CAM4-BAM and the experimental design. The modeling and





observational results are discussed in Sect. 3. Potential mechanisms of effects of the MAM TPSH on the East Asian dust cycle are shown in Sect. 4. Further discussion remarks and concluding remarks are shown in Sect. 5 and Sect. 6, respectively.

## 2 Description of data and model

### 2.1 Observational datasets

The data employed in this study is obtained from the regular surface meteorological observations from 73 meteorological stations of the China Meteorological Administration over TP (Wang et al., 2014; Duan et al., 2017). It includes historical four times daily observations of ground surface temperature, surface air temperature, and wind speeds at 10 m above the surface from 1980-2008, mainly over the central and eastern TP. The surface sensible heating flux is obtained from the above three meteorological parameters by the bulk aerodynamic method (Duan et al., 2011). The observed TPSH index used here is directly
derived from Figure 1b (Duan et al., 2017), defined as the time series of the standard anomalies of the MAM mean sensible heating flux averaged from these 73 meteorological stations during 1980-2008 with the linear trend excluded.

### 2.2 Reanalysis data

Note that the Modern-Era Retrospective Analysis for Research and Applications, version 2 (MERRA-2), is the first long-term global reanalysis for the satellite era after the year of 1980, which is based on the Goddard Earth Observing System version 5
model (GEOS-5), described by Randles et al. (2016). MERRA-2 makes some improvements in online aerosol fields interacting with the model radiative fields (Randles et al., 2016; Gelaro et al., 2017), compared to its predecessor MERRA-1 (Rienecker et al. 2011). It is shown that MERRA-2 aerosol vertical structure and surface particulate matter compare well with available satellite, aircraft, and ground-based observations including dust aerosols (Buchard et al., 2016; 2018; Randles et al., 2018; Song et al., 2018). Here, we used the MERRA-2 surface dust concentration of the years from 1980 to 2008 to examine the
relationship with the observed TPSH index.

### 2.3 Model and numerical experiments

The Community Atmosphere Model version 4 (CAM4) was released as atmospheric component of the Community Climate System Model version 4 coupled with a bulk aerosol model parameterization (BAM), which is labeled by CAM4-BAM (Neale et al., 2010). The model adopted externally mixed parameterizations and fixed size distribution of aerosols including sulfate,
dust, organic carbon (OC), black carbon (BC), and sea salt, detailedly described by Tie et al. (2005). The parameterized scheme of dust physical processes including emission, transport, and dry and wet deposition in the CAM4-BAM is derived from the Dust Entrainment And Deposition model (DEAD) from CAM3 with four dust size bins (Mahowald et al., 2006, 2014). This CAM4-BAM has been updated mainly form three aspects including a improved size distribution for dust emissions, updated dust optical properties of short-wave, and optimized soil erodibility maps to perform better in the simulation of global-scale
dust aerosols and its radiative properties (Albani et al., 2014). The snow-darkening processes induced by mineral dust and





black carbon were parameterized by the Snow, Ice, and Aerosol Radiative component (SNICAR) in CAM4-BAM, described by Flanner et al. (2007, 2009). Recently, this updated CAM4-BAM has been utilized to successfully study the dust cycle and its radiative feedbacks on the East Asian climate including dust direct effect and dust-in-snow effect (Xie et al., 2018a, 2018b; Shi et al., 2019).

The updated CAM4-BAM utilizes the finite-volume dynamical core (fvcore) with $0.9° \times 1.25°$ in the horizontal resolution and 26 levels in the vertical resolution in this work. The numerical experiment was conducted with the present-day climatological mean sea-ice concentrations, and sea surface temperature (SST), as well as present-day greenhouse gases during the simulated period. The present-day sea-ice concentration and SST were merged from the Hadley Centre Sea Ice and SST data set and the optimum interpolation SST data set (Rayner et al., 2003; Hurrell et al., 2008). A numerical experiment was integrated
over 37 years with 7 years for spin up, including the aerosol direct effect and snow-darkening effect of absorbing aerosols. It is noted that anthropogenic aerosols affect the monsoon atmospheric circulation and the summer surface precipitation over East Asia by aerosol direct and indirect effects (Liu et al., 2011; Wang et al., 2015; Xie et al., 2016a, 2016b), which may affect the regional emissions over East Asia. In order to reduce the anthropogenic aerosols' influence, the the anthropogenic aerosol and precursor gas emissions were fixed at the preindustrial day (PI) in the model simulation (Lamarque et al., 2010), using
emissions of the year 1850. The simulated results from the last 30 years are used to investigate the modulation of the MAM TPSH on the interannual variability of East Asian dust cycle, also compared with the observed results.

## 3 Results

### 3.1 Model evaluation against measurements

To evaluate the performance of CAM4-BAM, the simulated TPSH and dust concentration are compared with observed results of
the 73 meteorological stations and the MERRA-2 reanalysis data, especially in MAM. Figure 1 presents the spatial distribution of the climatological mean TPSH (a), standard derivation of TPSH (b), and the relative standard deviation of TPSH in MAM (c) calculated for 30 model years. It is noted that the simulated TPSH index is defined as the time series of the standard anomalies of the MAM surface sensible heating flux averaged over the TP region with height above 2500 m MSL for the 30 model years. Figure 1a shows the simulated TPSH has high values over the central and eastern TP ranging from 20 W m$^{-2}$ up to 70 W m$^{-2}$,
and low values over the western TP due mainly to spatial variabilities of snow cover over this region (Xie et al., 2018b). The spatial patterns of the MAM TPSH are basically consistent with the ground measurements, reanalysis, and satellite data (Shi et al., 2014). It presents the standard deviation (SD) in Figure 1b and relative standard deviation (RSD) of the MAM TPSH in Figure 1c for the 30 model years, respectively. The spatial pattern of SD is similar with its climatological mean pattern, with higher values of SD associated with larger TPSH over the central and eastern TP. The largest values of SD exceed 14 W
m$^{-2}$ (Figure 1b) over these regions, which leads to the largest values of RSD exceed 90% (Figure 1c). The result indicates a significant interannual variation of the MAM TPSH, which will affect the East Asian dust cycle through altering the regional large-scale atmospheric circulation.





Figure 1d shows the annual cycle of the simulated mean TPSH over the TP region with height above 2500 m MSL and the observed averaged TPSH over the 73 stations of the central and eastern TP (Duan et al., 2017). The observational result shows significant seasonal variations of the TPSH, reaching the highest value in spring and the lowest value in winter. This phenomenon of seasonal TPSH variation manifests itself in CAM4-BAM simulations, suggesting that CAM4-BAM can capture

the observed TPSH seasonal cycle reasonably well. However, there exists a large discrepancy between the simulated and observational TPSH values, due probably to the lack of measurement sites over the western TP and the model's coarse horizonal resolution.

For dust aerosols, we (Xie et al., 2018a) have shown that the updated CAM4-BAM has the ability to capture spatial distributions of the dust AOD and the surface dust concentrations over East Asia, showing strong and positive correlations with

observational sites in various regions of China on seasonal and annual means. Here, we further compare the spatial distribution of the simulated surface dust concentration with that of the MERRA-2 data in MAM in Figure 2. It is shown that the centers of high surface dust concentrations are located mainly in the Taklamakan and Gobi deserts in MERRA-2, where the dust concentrations are larger than 200 $\mu$g m$^{-3}$. High surface dust concentrations over western China and northern China are absolutely consistent with the numerical and observational results (e.g., Sun et al., 2001; Liu et al., 2004; Gong et al., 2006;

Zhao et al., 2006; Zhang et al., 2012; Sun et al, 2012; Lou et al., 2016). Large amounts of mineral dusts emitted from these two dust source regions are transported to downstream land and ocean regions through the westerly winds, e.g., the eastern China and northwestern Pacific. This is confirmed by the significant reduction of dust concentration from the western China to the northwestern Pacific (Figure 2a). The simulated result exhibits similar spatial patterns of surface dust concentration compared with MERRA-2, showing that surface dust concentration is significantly reduced from the dust source regions ($\sim$10$^2$ $\mu$gm$^{-3}$)

to the northwestern Pacific ($\sim$10$^0$ $\mu$gm$^{-3}$) in Figure 2b. Overall, the comparison shows that the model can successfully derive the spatial distributions of the MAM surface dust concentration, compared with MERRA-2.

## 3.2 Relationships between TPSH and surface dust concentration

In order to study the interannual variability of the MAM TPSH and its impacts on the dust aerosols over East Asia and over the downstream Pacific regions, Figure 3 shows the correlation coefficients between the MAM TPSH and surface dust con-

centration for the CAM4-BAM simulation and observations. As shown in Figure 3a, the anomalies of simulated surface dust concentration correlate positively with the simulated TPSH index over East Asia and the corresponding downwind regions. The correlation coefficients are larger over the East Asian dust source region (EA, 76$-$110° E and 30$-$43° N) and the northwestern Pacific (NP, 140$-$162° E and 35$-$45° N), statistically significant with 95% confidence. The regionally averaged correlation coefficients in EA and NP are 0.49 and 0.44, respectively (Figures 3b and 3c). A similar spatial pattern of correlation coefficients

is shown between the observed TPSH index (Duan et al., 2017) and the anomalies of the MERRA-2 surface dust concentration in Figure 3d. Furthermore, Figures 3e and 3f show the positive correlation coefficients of 0.52 over EA and 0.11 over NP, respectively. In general, both the model and the observations display significant positive correlations between the TPSH index and surface dust concentration, showing that the variation of MAM TPSH plays a significant role in spatial distribution of the surface dust concentration over EA and NP.





To further quantify the impacts of the MAM TPSH on dust concentration over EA and NP, Figure 4 shows the composite absolute and relative differences of surface dust concentrations between the strongest and weakest TPSH years for the model and MERRA-2, respectively. The strongest (weakest) years of the TPSH are defined as the indices larger (smaller) than one standard deviation. Six strongest and five weakest TPSH years in the model (Figure 3b) and four strongest and four weakest TPSH years in the measurements (Figures 3e) are identified. The simulated results show that the surface dust concentrations during the strongest TPSH years are higher over East Asia and the downwind regions in Figure 4a, with a maximum exceeding 50 $\mu g \ m^{-3}$ over the Taklamakan and Gobi deserts, compared with the weakest TPSH years. It shows the larger percentage changes over the study regions EA and NP during the strongest TPSH years in Figure 4b. Furthermore, the averaged surface dust concentrations during the strongest TPSH years in Table 1 are increased with 13.83 $\mu g \ m^{-3}$ (13.1%) over EA and 1.37 $\mu g \ m^{-3}$ (36.9%) over NP, respectively. The MERRA-2 also presents the similar spatial distributions of the increase in surface dust concentration during the strongest TPSH years over East Asia and the the downwind regions in Figures 4c and 4d, with an increase of 21.54 $\mu g \ m^{-3}$ (22.9%) over EA and 0.72 $\mu g \ m^{-3}$ (13.3%) over NP from Table 1. Therefore, compared to the weakest TPSH years, the MAM surface dust concentrations in the strongest TPSH years are much higher over EA and NP in both the model and MERRA-2 data (Figure 4).

Additionally, Figure 5 shows the simulated vertical profiles of the MAM dust concentration for the mean, strongest and weakest TPSH years over EA and NP. It presents that mean vertical dust concentrations over EA are much larger than that over NP similarly with the surface dust concentration (Figure 2). The vertical dust concentrations in the strongest TPSH years are also higher than that in the weakest TPSH years over EA (Figure 5a) and NP (Figure 5b), respectively.

### 3.3 Differences in dust cycle between strongest and weakest TPSH years

In this subsection, we compare differences in the dust cycle including dust loading, emissions, transport, dry and wet depositions between the strongest and weakest TPSH years in Figure 6, also shown in Table 2 for the corresponding averaged values over EA and NP. Figure 6a presents spatial distributions of the MAM mean dust loading from the model with higher dust loading over EA and lower loading over NP, which is very similar with that of the dust surface concentration from the model and the MERRA-2 data in Figure 2. Figures 6b and 6c show absolute and relative differences in dust loading between the strongest and weakest TPSH years over EA and NP. It shows that, compared to the weakest TPSH years, the dust loading can be significantly increased in the strongest TPSH years, with an increase of 33.72 mg m$^{-2}$ (16.9%) over EA and 11.72 mg m$^{-2}$ (28.5%) as shown in Table 2.

Figure 6d presents the spatial distributions of the MAM mean dust emissions over East Asia, exhibiting a maximum dust emission up to 100 g m$^{-2}$ per season over the Taklamakan desert and Gobi desert. It shows that dust emissions are remarkably increased over the dust source region in the strongest TPSH years, compared to the weakest TPSH years in Figures 6e and 6f. For the averaged value over EA, the MAM dust emissions in the strongest TPSH years are increased by 4.82 g m$^{-2}$ per season with the percentage of 29.2% in Table 2. The enhanced dust emissions over EA in the strongest TPSH years are mainly due to the increased surface wind speeds over this region (discussed in Section 4). Figure 6g presents the spatial distribution of the MAM mean vertically integrated horizontal dust fluxes (defined as dust transport), showing that the emitted dusts from EA





are transported to the downwind NP region through the westerly winds. The absolute and relative differences in dust transport between the strongest and weakest TPSH years show that the dust transports are enhanced by the increased TPSH over EA and NP in Figures 6h and 6i. The averaged increases of dust transports in the strongest TPSH years are 0.51 g m$^{-1}$ s$^{-1}$ with 46.2% over EA and 0.25 g m$^{-1}$ s$^{-1}$ with 41.3% over NP, respectively. The enhanced dust transports result mainly from the increased

dust emissions over the dust source and the increased westerly winds induced by the higher TPSH. Compared to the weakest TPSH years, dust dry and wet depositions in the strongest TPSH years are also increased over EA and NP in Figures 6k, 6l, 6n and 6o, also as shown in Table 2. Hence, the dust cycle-related processes (e.g., dust loading, dust emissions, dust transport, and dust dry and wet depositions) are all enhanced by the increased TPSH.

## 4    Potential mechanisms of TPSH effects on dust cycle

The TPSH in MAM directly affects the large-scale circulation through altering atmospheric heat source of the TP (Ye and Wu, 1998; Liu and Dong, 2013; An et al., 2015; Duan et al., 2017), and further influence the regional dust cycle over East Asia. In this section, we compare the differences in the regional climate between the strongest and weakest TPSH years to interpret the changes in the dust cycle by the increased TPSH as mentioned in Section 3.

     To illustrate the effect of TPSH on the regional climate, we first show the changes in the surface air temperature and

the corresponding energy balance at the surface in MAM. Figure 7 presents the spatial distribution of the MAM composite differences between the strongest and weakest TPSH years for the model in surface temperature, surface sensible and latent fluxes, as well as surface radiation forcing. Figure 7a shows that the surface air temperature in the strongest TPSH years is remarkably increased over TP, whereas the surface air temperature is reduced over the northern regions of TP. According to the spatial distribution of the surface sensible flux (Figures 7b), the surface latent flux (Figures 7c), and the surface radiative forcing

(Figure 7d), the increased surface temperature over TP (Figure 7a) is associated mainly with the significant increase in TPSH. The surface latent flux over TP is slightly negative in Figure 7c, whereas the surface radiative forcing over TP is positive, but much smaller than the surface sensible flux. Hence, it shows that the increased surface temperature over TP resulted mainly from the surface sensible heating flux. Additionally, the reduced surface air temperature over the northern regions of TP is more likely due to the horizontal temperature advection, which is similar with the atmospheric temperature at 500 hPa in following

paragraph.

     The spatial distribution of atmospheric temperature anomaly at 500 hPa between the strongest and weakest TPSH years is shown in Figure 8a. It shows significant increases in atmospheric temperature at 500 hPa in the strongest TPSH years over TP and the downstream Pacific region, where the maximum temperature anomalies exceed 1.5 °C and 0.9 °C over these two regions, respectively. The temperature anomaly at 500 hPa over TP is mainly resulted from the increased diabatic heating over

TP in Figure 8b induced by the surface sensible flux (Figure 7b). The atmospheric temperature anomaly at 500 hPa over the Pacific region is mainly due to the diabatic heating in Figure 8b and the horizontal temperature advection in Figure 8d. It is noted that the decrease in atmospheric temperature at 500 hPa over the northern regions of the TP (Figure 8a) is resulted from the horizontal temperature advection in Figure 8d. The increases in atmospheric temperature in the strongest TPSH years over





TP and the downstream Pacific region enhance the geopotential height and produce two anticyclonic anomalies at 500 hPa over these two regions (Figure 9a). In the high-level atmosphere of 200 hPa, the increased geopotential height and the two anticyclonic anomalies induced by the stronger TPSH are also shown over the two regions in Figure 9b, similarly with those at 500 hPa (Figure 9a). The atmospheric circulation anomalies induced by TPSH are absolutely same as the results from the

sensitivity of TPSH in GCM in the previous investigation (Duan et al., 2017). These two anticyclonic anomalies over TP and the downstream Pacific region at middle and high levels, significantly enhance the westerly winds over EA and NP, as shown in Figure 9. Furthermore, we show the height-latitude cross-section of the MAM composite difference between the strongest and weakest TPSH years in atmospheric temperature and zonal winds over EA and NP in Figure 10. The higher TPSH in the strongest TPSH years heats the surface, extending to the middle and high level atmosphere over TP and increasing the westerly

winds from low to high levels over EA ($30-43°$ N) in Figure 10a. The atmospheric temperature over NP are significantly increased at middle level and the westerly winds from low to high levels are enhanced over this region ($35-45°$ N) in Figure 10b. Hence, the higher TPSH induces the two anticyclonic anomalies at middle and high level over TP and the Pacific region (Figure 9), significantly enhancing the westerly winds over EA and NP (Figures 9 and 10). These increased westerly winds in the strongest TPSH years over EA and NP can enhance the dust transport from the dust source regions. Additionally, the

increased westerly winds over the EA induced by the higher TPSH also increase the surface 10-m wind speeds in Figure 11, which further enhance the dust emissions over EA (Figures 6e and 6f).

## 5   Further discussion

On one hand, an increase in dust aerosols would result in a decrease of solar heating at surface, and thus a decrease in temperature, turbulence, and thus sensible heat flux (Miller et al., 2004; Heinold et al., 2007; Xie et al., 2018a). On the other hand,

previous studies indicate that dusts deposited on the snow over the TP region reduce the visible surface albedo and warm the surface of TP through the snow-darkening effect (Flanner et al., 2009; Qian et al., 2011, 2015; He et al., 2018; Xie et al., 2018; Shi et al., 2019). The dust-in-snow effect over TP increases the TP's thermal effects through enhancing the heat fluxes at the surface, and then enhances the westerly winds and dust cycle over East Asia (Xie et al., 2018b). A significant feature of dust-in-snow effect over the TP creates a positive feedback loop enhancing the East Asian dust cycle. Our results show that,

compared to the weakest TPSH years, the MAM dust cycle in the strongest TPSH years are much more vigorous over East Asia. In the strongest TPSH years, much more dusts deposited on snow over TP will further enhance the regional dust cycle through the above positive feedback loop of dust-in-snow. Hence, we believe that the positive feedback of dust-in-snow over TP plays a non-neglectable role in modulation of surface sensible heating over TP on the interannual variability of East Asian dust cycle.

Note that surface fluxes including sensible heat, latent heat, and surface momentum are parameterized by resolved meteorological quantities (e.g., ground temperature, surface air temperature, wind speed) in CAM4-BAM (Neale et al., 2010), and the performance of the parameterizations strongly depends on these input variables (Liu et al., 2013). The coarse horizonal resolution can not resolve more details of the TP's complex topography, which results in inadequate meteorological quantities





over this region (Li et al., 2015; Ramu et al., 2016) and the discrepancy between the simulated and observational TPSH values (Figure 1d). Hence, the simulation with higher horizonal resolution makes some significant improvements in the simulated meteorological quantities over TP (Li et al., 2015; Ramu et al., 2016) and in turn TPSH, which can evaluate exactly the impact of TPSH on the East Asian dust aerosols.

## 6 Concluding remarks

It is well known that the surface sensible heating in MAM (March-April-May) over the Tibetan Plateau (TPSH) can affect the Asian hydrological cycle and the regional large-scale circulation through altering the atmospheric heat source of TP (Wu and Zhang, 1998; Ye and Wu, 1998; Duan et al., 2011, 2017; Wu et al., 2012; An et al., 2015). It is likely that the changes in the large-scale atmospheric circulation induced by the MAM TPSH influence the East Asian dust cycle by altering the dust

emissions and dust transports. In this study, we firstly investigated the impact of the MAM TPSH on the interannual variability of East Asian dust cycle based on the CAM4-BAM, the MERRA-2 surface dust concentration, and TPSH measurements.

The results of CAM4-BAM shows the dust surface concentrations in MAM are significantly positively correlated with the TPSH over the East Asian dust source (EA) and over the northwestern Pacific (NP), with regionally averaged correlation coefficients of 0.49 for EA and 0.44 for NP. Similar positive correlations are also shown to exist between the MAM TPSH

measurements averaged over the 73 observation sites and the MERRA-2 surface dust concentration with 0.52 for EA and 0.11 for NP. Further comparisons between the strongest and weakest TPSH years reveal that, the MAM dust surface concentration in the strongest TPSH years increases with relative differences of 13.1% in model (22.9% in MERRA-2) over EA and 36.9% in model (13.3% in MERRA-2) over NP, respectively. The simulated results show that the processes of whole dust cycles including dust loading, emission, and transport, as well as dust depositions are also significantly enhanced during the strongest

TPSH yeas over EA and NP. Through enhancing the atmospheric heat source of TP, stronger TPSH in MAM generates an anticyclonic anomaly in the middle and upper levels over TP and over the downstream Pacific region, respectively. These two induced anticyclonic anomalies induced by the increased TPSH result in increasing the westerly winds over both EA and NP, which then enhances dust emissions over the dust source regions, and dust transports over these two regions, as well as the regional dust cycles.

Increasing greenhouse gas can significantly influence the global and regional dust cycle through affecting the climate, vegetation and dust-climate feedbacks although there exists large uncertainty in the response of the global dust loading to future climate change (Harrison et al., 2001; Tegen et al., 2004; Mahowald et al., 2006; Kok et al., 2018). Additionally, the surface sensible flux over TP exhibits a remarkable weakening trend under global warming, induced primarily by weakened surface wind speeds (Duan et al., 2011; Wang et al., 2012; Yang et al., 2014). Our study shows the importance of the TPSH on the

change in East Asian dust cycle through affecting the westerly winds. Therefore, these results suggest that addressing the East Asian dust changes in the future climates require considering not only increasing greenhouse gas emissions but also the variations of the TP's heat source under global warming.





*Data availability.* Data of CAM-BAM are available upon request from the corresponding author (xnxie@ieecas.cn) for access. The MERRA-2 reanalysis data are developed by the GMAO with support from the NASA Modeling, Analysis and Prediction program, acquired from https://goldsmr5.gesdisc.eosdis.nasa.gov/data/ (last access: 14 Sept. 2018).

*Author contribution.* XX, XL, and ZS designed the numerical experiments and XX performed all the simulations. XX

5   prepared the paper with substantial contribution from all co-authors.

*Competing interests.* The authors declare that they have no conflict of interest.

*Acknowledgements.* This study was partially funded by the Strategic Priority Research Program of Chinese Academy of Sciences (Grant No. XDA20070103), the National Key Research and Development Program of China (Grant No. 2016YFA0601904), and the National Natural Science Foundation of China (Grant No. 41690115). Z. Shi is the support of Youth Innovation Promotion Association CAS. Y. Liu

10   is supported by the US Department of Energy's Atmospheric System Research (ASR) program.



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





**Table 1.** The March-April-May (MAM) averaged surface dust concentration ($\mu$g m$^{-3}$) over the East Asian dust source area (EA, $76-110^\circ$ E and $30-43^\circ$ N) and over the northwestern Pacific (NP, $140-162^\circ$ E and $35-45^\circ$ N) from the model and the MERRA-2 reanalysis in the Mean, Strongest, and Weakest years, as well as the absolute difference (Strongest$-$Weakest) and the relative difference ((Strongest$-$Weakest)/Weakest$\times100\%$).

| Regions | Mean | Strongest | Weakest | Diff. | Diff. (%) |
|---|---|---|---|---|---|
| EA (Model) | 107.12 | 119.33 | 105.50 | 13.83 | 13.1% |
| NP (Model) | 3.87 | 5.08 | 3.71 | 1.37 | 36.9% |
| EA (MERRA-2) | 114.15 | 115.64 | 94.10 | 21.54 | 22.9% |
| NP (MERRA-2) | 5.99 | 6.14 | 5.42 | 0.72 | 13.3% |





**Table 2.** The March-April-May (MAM) dust cycle including dust laoding (mg m$^{-2}$), dust emission (g m$^{-2}$ per season), dust transport (g m$^{-1}$ s$^{-1}$), dust dry deposition (g m$^{-2}$ per season), and dust wet deposition (g m$^{-2}$ per season) over the East Asian dust source region (EA, $76-110°$ E and $30-43°$ N) and over the northwestern Pacific (NP, $140-162°$ E and $35-45°$ N) from the model in the Mean, Strongest, and Weakest years, as well as the absolute difference (Strongest−Weakest) and the relative difference ((Strongest−Weakest)/Weakest×100%).

| Regions | Dust cycle | Mean | Strongest | Weakest | Diff. | Diff. (%) |
|---------|-----------|------|-----------|---------|-------|-----------|
| EA | Dust loading | 204.65 | 233.32 | 199.60 | 33.72 | 16.9% |
|    | Emission | 17.93 | 21.31 | 16.49 | 4.82 | 29.2% |
|    | Transport | 1.26 | 1.63 | 1.11 | 0.51 | 46.2% |
|    | Dry deposition | 9.16 | 11.17 | 8.50 | 2.67 | 31.4% |
|    | Wet deposition | 5.00 | 5.27 | 4.69 | 0.58 | 12.4% |
| NP | Dust loading | 42.94 | 52.91 | 41.19 | 11.72 | 28.5% |
|    | Emission | − | − | − | − | − |
|    | Transport | 0.67 | 0.85 | 0.60 | 0.25 | 41.3% |
|    | Dry deposition | 0.18 | 0.24 | 0.17 | 0.07 | 42.4% |
|    | Wet deposition | 0.89 | 1.01 | 0.86 | 0.16 | 18.5% |

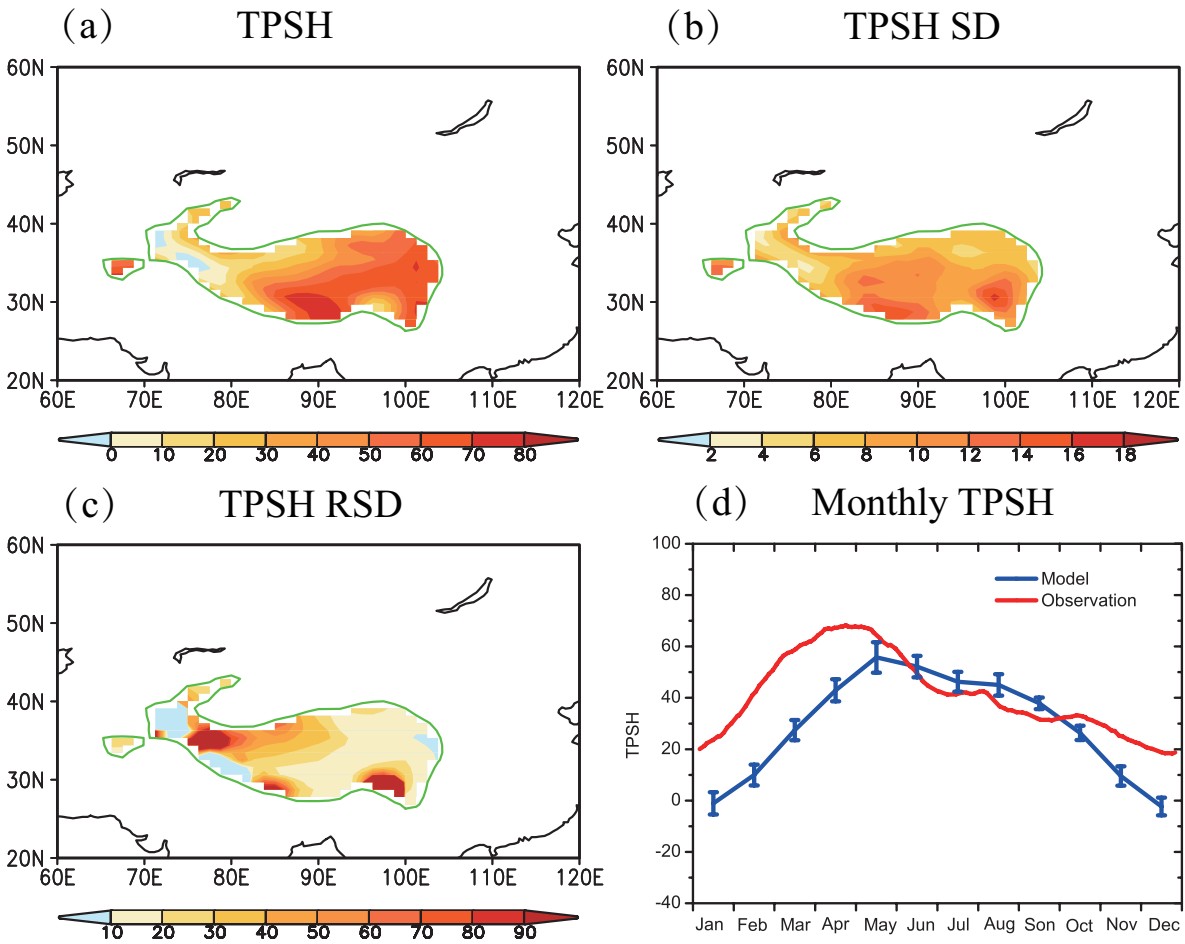

**Figure 1.** Spatial distribution of (a) the MAM mean sensible heat over the TP (TPSH, W m$^{-2}$), (b) the standard derivation of TPSH (SD, W m$^{-2}$), and the relative standard deviation of TPSH (RSD, %) in MAM calculated for 30 model years. (d) Annual cycle of TPSH from the model averaged over the TP region above 2500m and the observations averaged over the 73 stations over the TP (Duan et al., 2017). Note that the error bars (d) represent the standard deviation of the simulated TPSH for 30 model years. The green-contour area indicates the plateau above 2500 m.



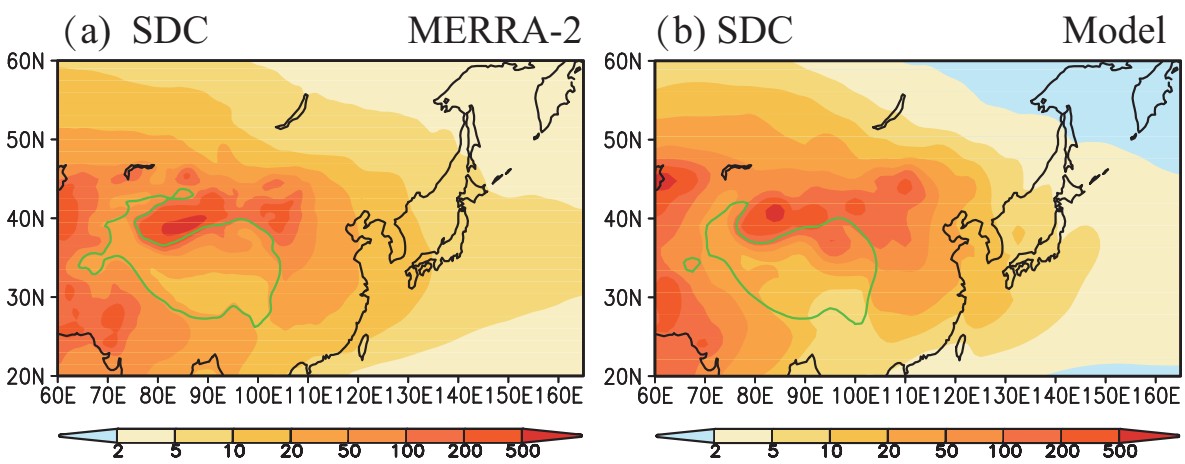

**Figure 2.** (a) Spatial distribution of the MAM mean surface dust concentration (SDC, $\mu$g m$^{-3}$) for the 30 model years and (b) for the years of 1980−2008 derived from the MERRA-2. The green-contour area indicates the plateau above 2500 m.

**Figure 3.** (a) Spatial distribution of the correlation coefficients between the index of sensible heat over the TP (TPSH index) and the anomalies of surface dust concentration in MAM for the 30 year CAM4-BAM simulation and (b) the observed TPSH index (Duan et al., 2017) and the anomalies of the MERRA-2 surface dust concentration for the years of 1980−2008, and the corresponding time series of regionally averaged surface dust concentration (red line) over East Asian dust source area (b for model and e for observations) and over northwestern Pacific (c for model and f for observations) and the TPSH index (blue bars). Note that the dots (a, d) represent the grid points where the changes pass the two-tailed t test at the 5% significance level and the green-contour area indicates the plateau above 2500 m.



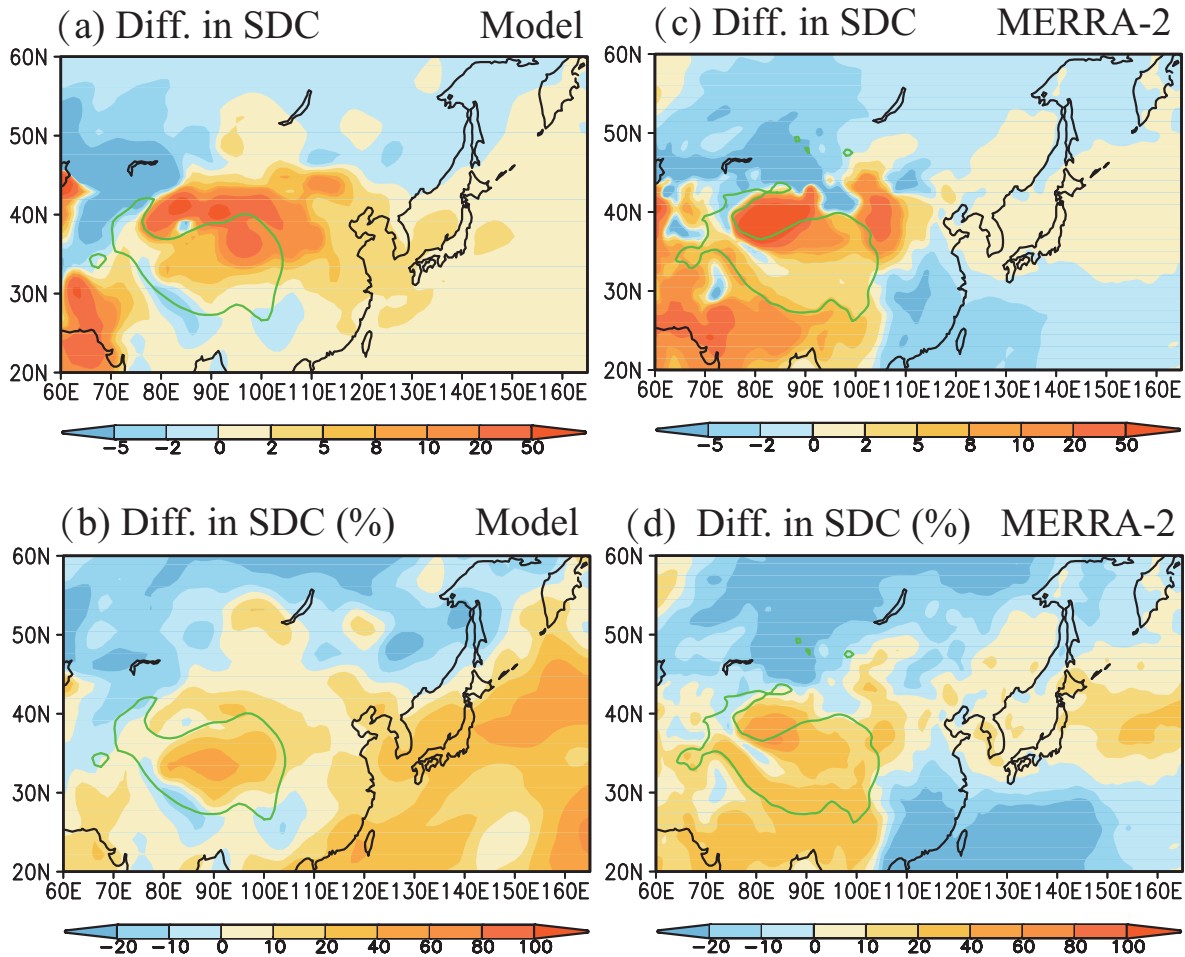

**Figure 4.** (a, c) Composite absolute difference of the MAM mean surface dust concentration (SDC, $\mu$g m$^{-3}$) and (b, d) the corresponding composite relative differences between the strongest and weakest TPSH years (strongest−weakest) for the model and the MERRA-2, respectively. The green-contour area indicates the plateau above 2500 m.

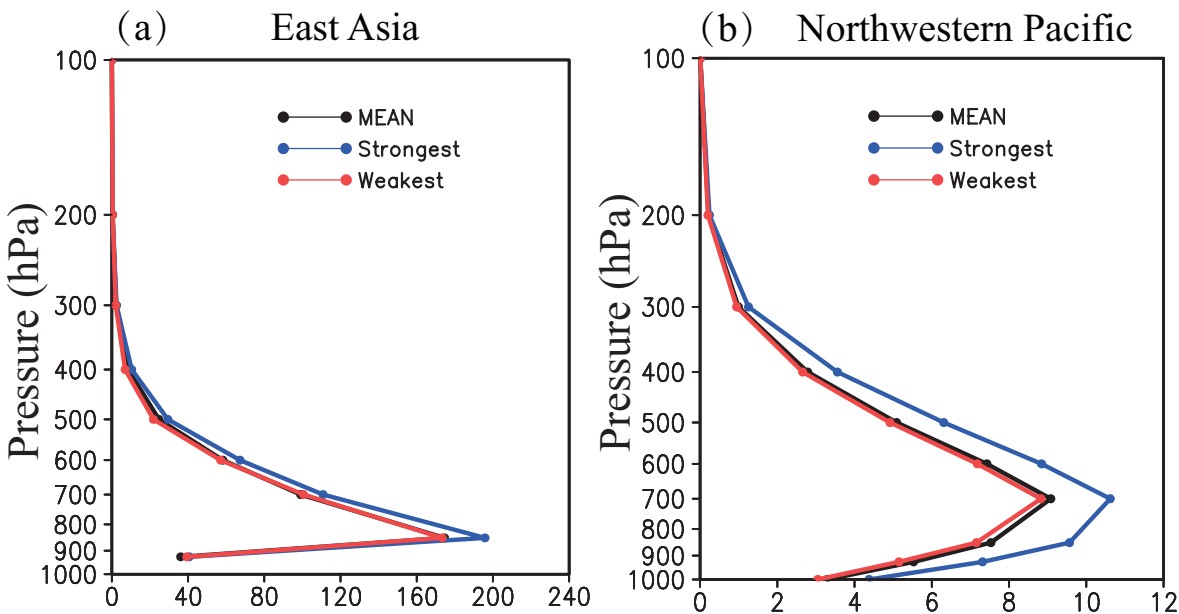

**Figure 5.** Simulated vertical profile of the MAM dust concentration (ug kg$^{-1}$) from the model for the mean, strongest and weakest years (a) over East Asian dust source area and (b) over the northwestern Pacific.



**Figure 6.** Spatial distribution of the MAM mean values for 30 model years (left column), (middle column) absolute differences between the strongest and weakest TPSH years (strongest−weakest), and (right column) the corresponding relative differences in (a, b and c) dust laoding (mg m$^{-2}$), (d, e and f) dust emissions (g m$^{-2}$ per season), (g, h and i) dust transport (g m$^{-1}$ s$^{-1}$), (j, k and l) dust dry deposition (g m$^{-2}$ per season), and (m, n and o) dust wet deposition (g m$^{-2}$ per season). The green-contour area indicates the plateau above 2500 m.

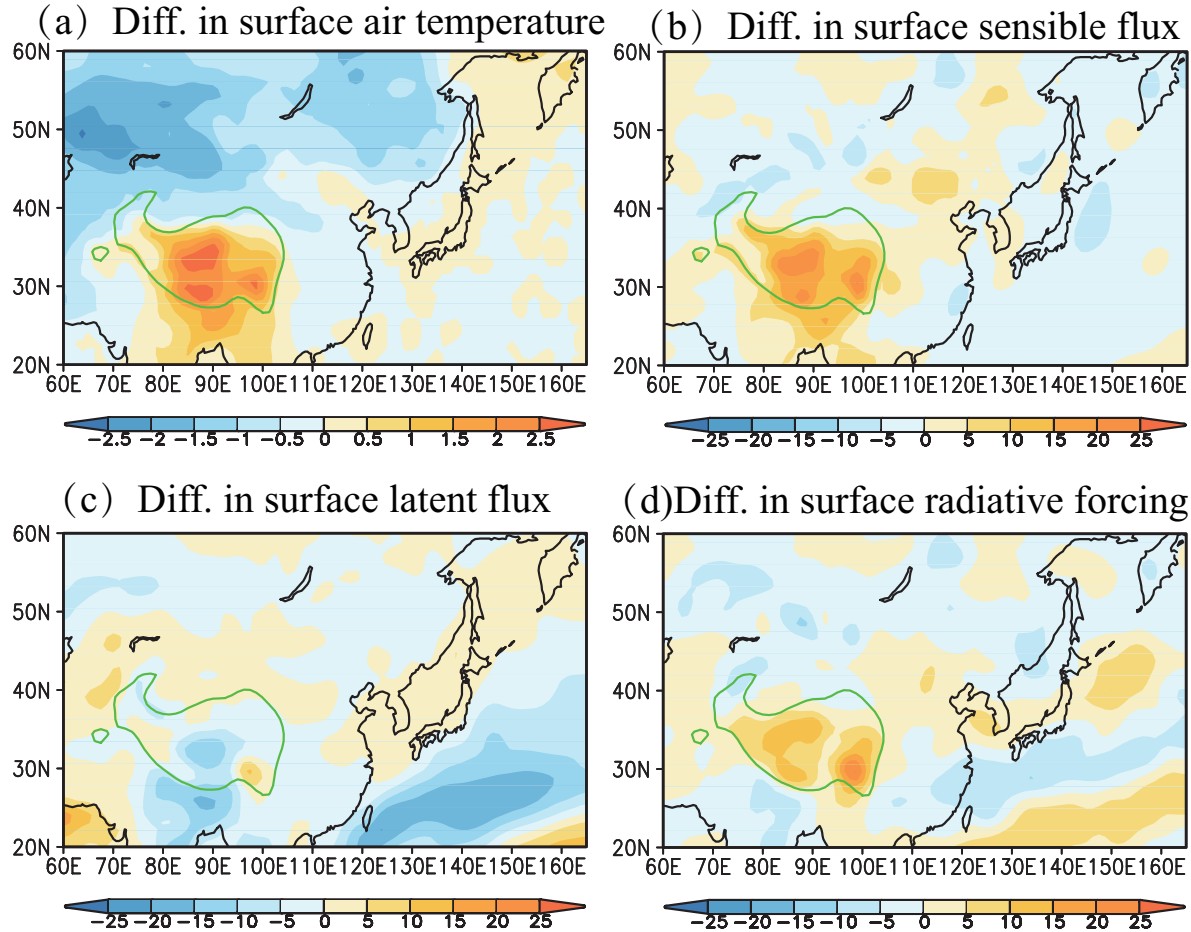

**Figure 7.** Spatial distribution of the MAM composite difference between the strongest and weakest TPSH years (strongest−weakest) for the model in (a) the surface air temperature (°C), (b) the surface sensible heat (W m$^{-2}$), (c) the surface latent heat (W m$^{-2}$), and (d) the surface radiative forcing (W m$^{-2}$). The green-contour area indicates the plateau above 2500 m.

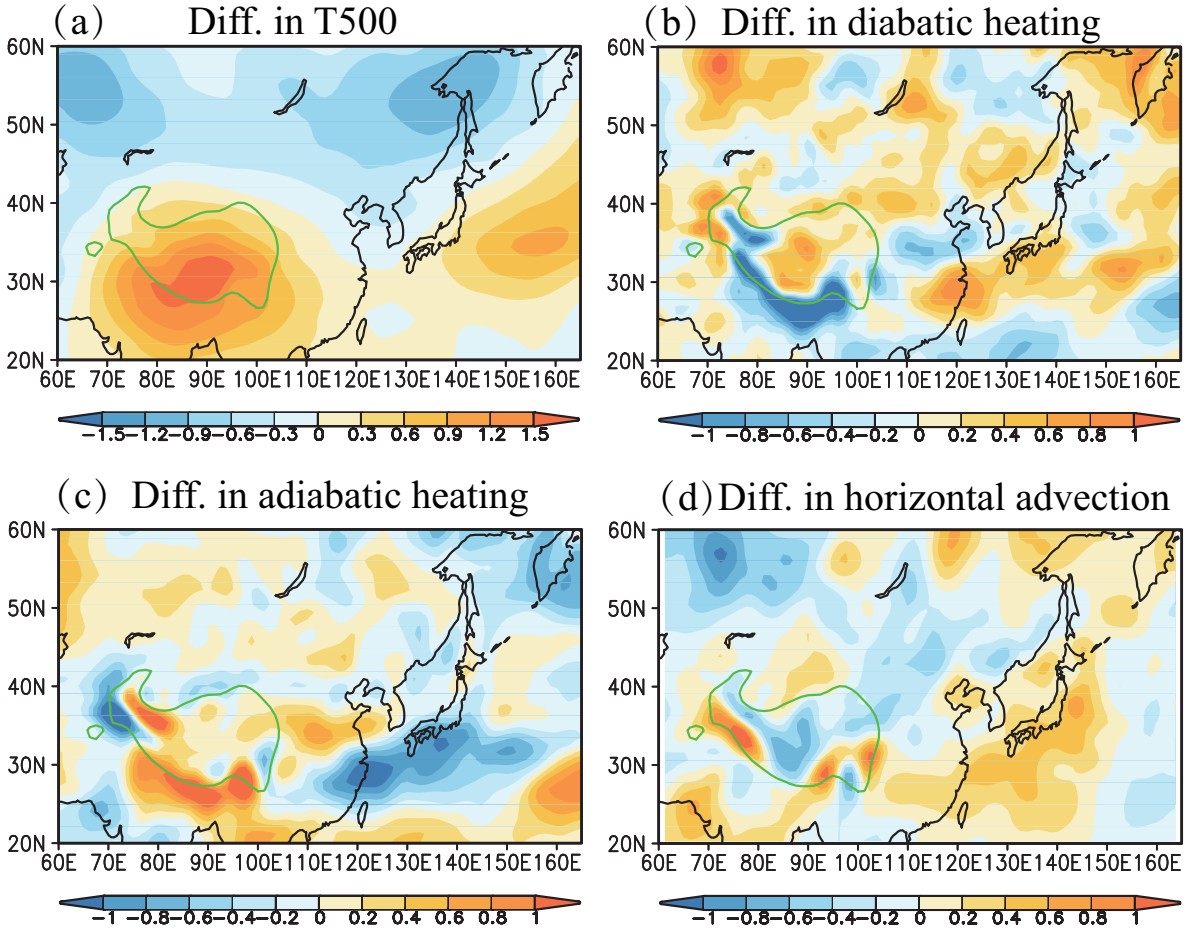

**Figure 8.** Spatial distribution of the MAM composite difference between the strongest and weakest TPSH years (strongest−weakest) for the model in (a) the atmospheric temperature at 500 hPa (°C), (b) the diabatic heating at 500 hPa (K day$^{-1}$), (c) the adiabatic heating at 500 hPa (K day$^{-1}$), and (d) the horizontal temperature advection at 500 hPa (K day$^{-1}$). The green-contour area indicates the plateau above 2500 m.

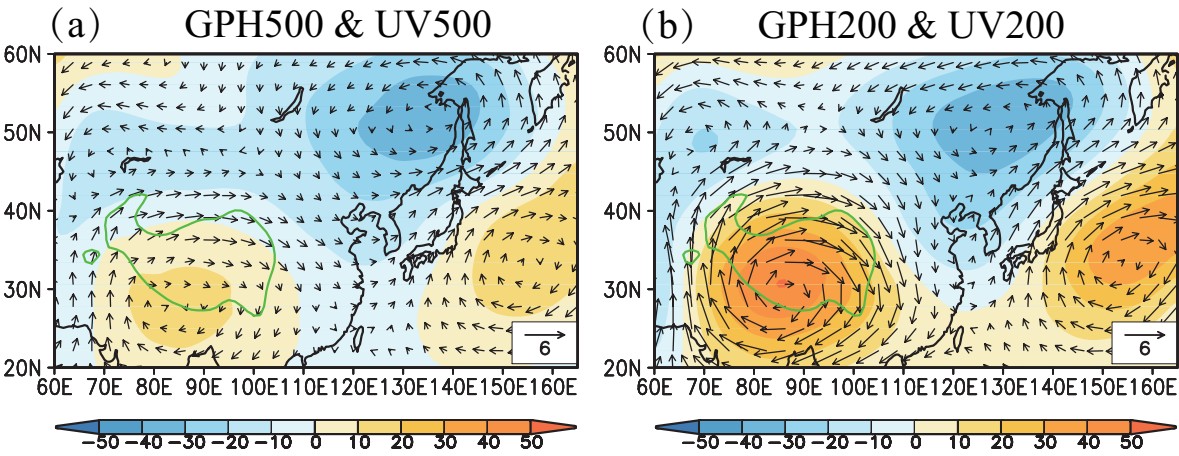

**Figure 9.** Spatial distribution of the MAM composite difference between the strongest and weakest TPSH years (strongest−weakest) for the model in geopotential height (GPH, gpm) and wind vectors (m s$^{-1}$) at (a) 500 hPa level and (b) 200 hPa level. The green-contour area indicates the plateau above 2500 m.



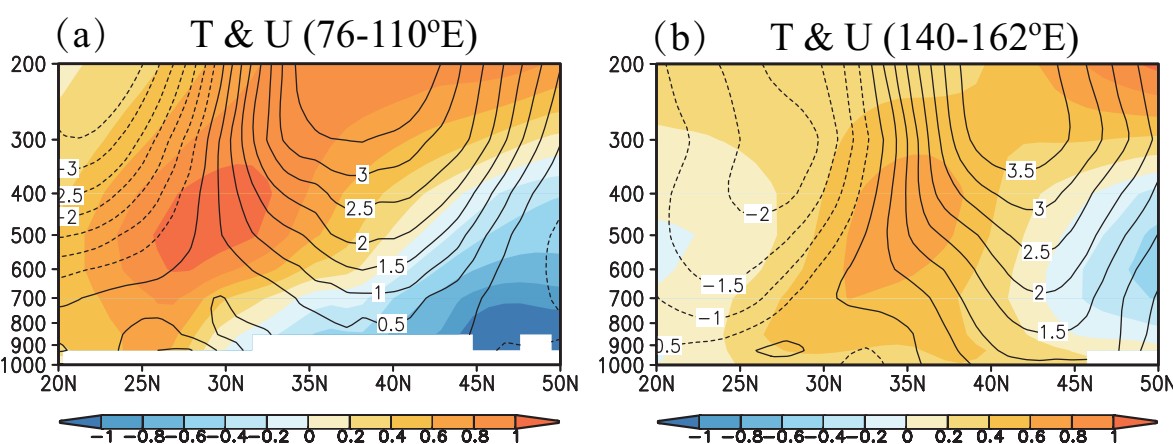

**Figure 10.** Height-latitude cross-section of the MAM composite difference between the strongest and weakest TPSH years (strongest−weakest) for the model in atmospheric temperature (shaded, °C) and zonal winds (contoured, m s$^{-1}$) along (a) 76−110° E and (b) 140−162° E.

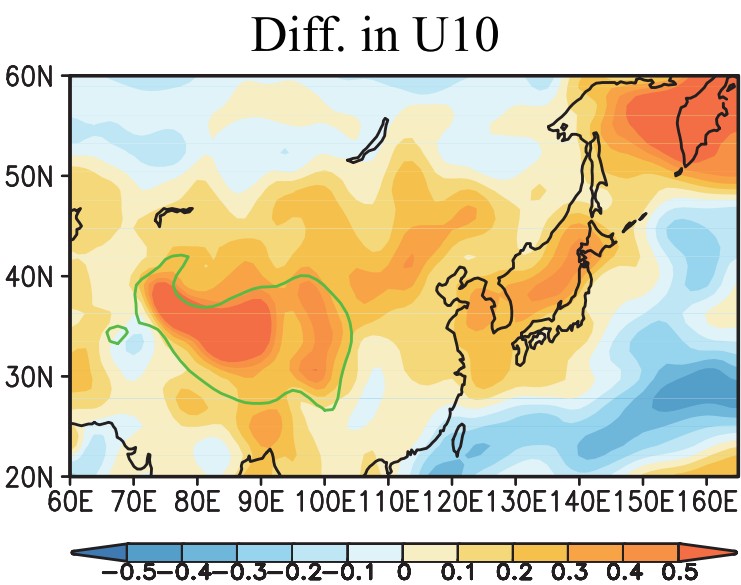

**Figure 11.** Spatial distribution of the MAM composite difference between the strongest and weakest TPSH years (strongest−weakest) for the model in 10-m wind speed (U10, m s$^{-1}$). The green-contour area indicates the plateau above 2500 m.