# Peer review of "Modulation of surface sensible heating over the Tibetan Plateau on the interannual variability of East Asian dust cycle"

_Atmospheric Chemistry and Physics, 2019_

## Referee Comment (RC1) · Anonymous Referee #2 · 21 Jun 2020

**Review for article " Modulation of surface sensible heating over the Tibetan Plateau on the interannual variability of East Asian dust cycle" by Xiaoning Xie and co-authors**

**Synopsis & General remarks:**

The authors present a research on the interannual variability of East Asia dust cycles by using the model CAM4-BAM, the surface meteorological data in TP and the global reanalysis from MERRA-2 as observations. They have found the interesting relations between the dust cycle variability in East Asia and the MAM surface sensible heating from Tibetan Plateau (TPSH) which has triggered two anticyclonic anomalies in the dust source area and in the downwind North Pacific Ocean. This kind of anomalous circulations finally enhances the surface wind at 10-m which is absolutely determined to enhance the dust emissions as the 10-m wind is the key input for dust emission in the dust product scheme MB95 used in CAM4-BAM. The manuscript is written in a well structure way and in fluent and concise English. The results are reasonable and interest. This work also gives us a clue to pin down the relations between the dust cycling and the regional climate in East Asia from the long time scale point of view, especially the importance role of the TP.

Taklimakan and the Gobi desert are the two major dust sources in East Asia. The dust is emitted mostly as the cold wind with the Siberian High comes through these two areas. As the cold air mass in East Asia is mostly dominated by the position of polar vortex in north Hemisphere and absolutely also influenced by the huge complex terrain of TP. So the authors should be careful to draw the conclusion the TP is the dominating factor for all the variabilities of the dust cycling in East Asia.

**Verdict**
Some major and some minor revisions have to be done. In particular, the authors need to be on alert about the conclusions they get.

**Major Comments:**
1.  Page 4 Line 9: As the TPSH index is very important in this manuscript, please add how you get it by the bulk aerodynamics method in detail to make it more readable.

2.  Page5 Line10: How do you set you simulation time period? Is it the same time period as 1980-2008 of MERRA-2 except the spin up time? Please clarify.

3.  Section 3.2: Researches have shown that the dust emission trend in East Asia is decreasing since 1970 but with a small increase trend from 1995 to 2003. The results in Figure 3f is consistently showed this small increase trend while the model results in Figure 3c didn't. This is also reasonable that the correlation coefficient in Figure3F is much smaller than that in Figure 3c. In this case, the authors should investigate and try to explain the difference with the information of other factors such as the activity of the

polar vortex, an important factor that influence the cold air mass activities then hence the dust emission activities in the targeting region.

4. Section 3.3:In figure 6d-6f, there are remarkable dust emissions in west and central TP, this not consistent with the normal dust sources distributions in East Asia in other researches. Maybe there are some errors in the source data, please clarify.

5. Section 4:    All the meteorological results related to the TPSH's potential mechanisms of the effects on dust cycle are deduced from the model result. Have you ever evaluated the basic parameters of temperature and wind with the routine meteorological measurements or the reanalysis in the targeting region? As the differences in temperature and in wind from Figure7-10 are quite small which may be in the same range of or be noised by the bias of the modelling.

6. Page9 Line28-29: As there is no detail evidence from the manuscript for the dust deposition on **SNOW** and so for the feedbacks, please be careful to draw the conclusion.

**Minor Comments:**

1. Page3 Line 34:    Is the 'CAM-BAM' right? Should it be 'CAM4-BAM'? Please check.

2. Page5 Line 27-28: For the sentence ' It presents …, 30 model years, respectively', it is repeated described, please delete it.

3. Page8 Line18-20: For the sentence 'According to … with the significant increase in TPSH', it is not logically right. Please find another way to describe it and make it more readable.

4. Page 19:    The 'C" is missing in the figure caption.

5. Page 21: Please use the same dust concentration scale for Figure 3b and Figure 3e, for Figure 3c and figure 3f, to make it more readable.

   Do you mean the dots in the black box? They are not very clear in Figure 3a and Figure 3d as the dots look like some lines. Please clarify.

---

## Referee Comment (RC2) · Anonymous Referee #1 · 26 Jun 2020

**Review Comments for "Modulation of surface sensible heating over the Tibetan Plateau on the interannual variability of East Asian dust cycle" by Xie et al.**

Dust has important effects on TP and Asian regional climate, while the change in TP climate can feed back to dust generation and lifecycle. The authors conducted model simulations combined with analysis of observations and reanalysis datasets to quantify the effects of TPSH on the interannual variability of dust life cycle in East Asia. This would potentially enhance our understanding of the interactions between dust aerosols and Asian/TP climate, and hence regional climate change. The manuscript is generally well-structured, and the methodology is also sound. Before it can be considered for publication, I have a few comments and suggestions to potentially improve the quality of the manuscript. Particularly, more clarifications and discussions are needed in the modeling and analysis parts. Please see my specific comments below.

Comments
1. Title: The authors mainly focused on MAM TPSH, so I suggest including "Springtime" in the title to avoid any confusion.

2. Introduction: It would be good if the authors could explicitly highlight the difference and novelty of this study compared with their previous study (Xie et al., 2018b), since there are some overlaps between the two studies.

3. Section 2: (1) The website links for the observation and modeling data used in this study need to be provided if they are publicly accessible. (2) For the model, I suggest the authors use CAM-chem/MAM in the future study, which includes more realistic aerosol representations. Besides, the assumption of aerosol external mixing and/or simplified (i.e., spherical) particle structure/morphology in CAM4-BAM can lead to uncertainty/bias in DRF (e.g., He et al., 2015: https://doi.org/10.5194/acp-15-11967-2015; Scarnato et al., 2015: https://doi.org/10.5194/acp-15-6913-2015). This issue needs to be discussed to some extent in the manuscript. (3) Was the aerosol-cloud interaction included in the model? The authors did not mention this. (4) I am not quite convinced that it is a good idea to fix the emissions in year 1850. As the authors mentioned, the anthropogenic emissions could serve as a confounding factor for the perturbation of the MAM TPSH impact. If removing the anthropogenic emissions, then the resulting effect of TPSH on dust cycle would not be realistic. If so, the authors need to make some clarifications in the title, abstract, and introduction sections to state that this study only considers the scenario without anthropogenic emission effect. This issue is particularly important when considering that the observations used to evaluate model actually reflect both the natural dust emissions and anthropogenic emissions of other species throughout the study period.

4. Section 3.1: (1) Some more discussions on the reasons of the spatial patterns and biases of TPSH need to be provided. Currently, the authors mainly described the results without many explanations of the physics behind in this section. (2) A particularly interesting question is that dust also feeds back to the TP climate and affect TPSH. For example, column dust leads to surface dimming and reduce TPSH, while dust in snow enhance surface warming and increase TPSH. So how does this dust feedback contribute to the variation of TPSH under the effects of other influential factors? The authors mainly discussed the effect of TPSH on dust in the

following sections, but it will be interesting to see the effect of dust on TPSH too. (3) The authors seem to focus on evaluating modeled surface dust concentrations only. Why not also evaluate the column property such as AOD? This at least will be meaningful over dust source regions. This is also important for the discussions on dust loading in the following sections.

5. Section 5: (1) The authors made the argument that the dust-in-snow feedback plays an important role, but without any quantitative analysis and/or figures. The dust-in-snow feedback is actually an interesting point, so it would be good to see some quantitative results, figures, and discussions to back up the argument. Besides, a recent study (He et al., 2019: https://doi.org/10.1029/2019MS001737) showed that the dust-snow albedo effect/feedback can be significantly enhanced by dust-snow internal mixing compared with external mixing (presumably assumed in CAM4-BAM model). This could potentially enhance the importance of the dust-in-snow feedback in modulating TPSH. Some discussions on this aspect would be useful. (2) Since a number of potential uncertainty factors (some of them have been mentioned in my comments above) could be involved in the model simulations and analysis, I suggest including a paragraph or two to specifically discuss the uncertainties of this study in this section.

---

## Author Comment (AC1) · 4 Aug 2020

Manuscript Number: acp-2019-393

Journal: ACP

The revised manuscript entitled "Modulation of springtime surface sensible heating over the Tibetan Plateau on the interannual variability of East Asian dust cycle" by Xiaoning Xie, Anmin Duan, Zhengguo Shi, Xinzhou Li, Hui Sun, Xiaodong Liu, Xugeng Cheng, Tianliang Zhao, Huizheng Che, and Yangang Liu

We thank the ACP Handing Editor (**Professor Kari Lehtinen**) for his hard work and the two anonymous referees for their constructive suggestions to improve our manuscript significantly. We greatly appreciate the generally positive comments from both the two Reviewers (Reviewer #1 and Reviewer #2), and have addressed all the concerns, with point-by-point responses detailed below (reviewers comments in red color and our responses in blue color). We have uploaded the file of "Response to reviewers.pdf".

Best wishes,

Xiaoning Xie

Response to Reviewer #2:

Synopsis & General remarks:

The authors present a research on the interannual variability of East Asia dust cycles by using the model CAM4-BAM, the surface meteorological data in TP and the global reanalysis from MERRA-2 as observations. They have found the interesting relations between the dust cycle variability in East Asia and the MAM surface sensible heating from Tibetan Plateau (TPSH) which has triggered two anticyclonic anomalies in the dust source area and in the downwind North Pacific Ocean. This kind of anomalous circulations finally enhances the surface wind at 10-m which is absolutely determined to enhance the dust emissions as the 10-m wind is the key input for dust emission in the dust product scheme MB95 used in CAM4-BAM. The manuscript is written in a well structure way and in fluent and concise English. The

results are reasonable and interest. This work also gives us a clue to pin down the relations between the dust cycling and the regional climate in East Asia from the long time scale point of view, especially the importance role of the TP.

Taklimakan and the Gobi desert are the two major dust sources in East Asia. The dust is emitted mostly as the cold wind with the Siberian High comes through these two areas. As the cold air mass in East Asia is mostly dominated by the position of polar vortex in north Hemisphere and absolutely also influenced by the huge complex terrain of TP. So the authors should be careful to draw the conclusion the TP is the dominating factor for all the variabilities of the dust cycling in East Asia.

Verdict. Some major and some minor revisions have to be done. In particular, the authors need to be on alert about the conclusions they get.

Response: Thank Reviewer #2 very much for the positive comments and constructive suggestions. Yes, I absolutely agree with the Reviewer' comments about dominating factor of dust storm activities. Observations show a significant decreasing trend form 1970 and an increase trend from 1995 to 2003 in dust activities. Hence, there exists a decadal change in dust activities, which is mainly related to polar vortex activities (Qian et al., 2002; Zhao et al., 2004; An et al., 2018). Hence, we believe that polar vortex activities have a dominated role in determining eastern Asian dust storms. Through our analysis of observations and model, we claimed that TPSH can modulate the interannual variability of the eastern Asian dust cycle, showing that is a non-negligible factor.

1. Page 4 Line 9: As the TPSH index is very important in this manuscript, please add how you get it by the bulk aerodynamics method in detail to make it more readable. Yes, we have added the corresponding description about the bulk aerodynamics method in page 4. The data includes historical four times daily observations of ground surface temperature ( $T_s$ ), surface air temperature ( $T_a$ ), and wind speeds at 10 m above the surface ( $V_{10m}$ ) from 1980-2008, mainly over the central and eastern TP. The surface sensible heating flux (SH) is obtained from the above three meteorological parameters by the bulk aerodynamic method (Duan et al., 2011; 2017), which is expressed as follows,

 $SH = C_p \ \rho \ C_{DH} \ V_{10m} \ (T_s - T_a),$

where  $C_p$  is the specific heat of dry air at constant pressure ( $C_p=1005 \text{ J kg}^{-1}\text{K}^{-1}$ ),  $\rho$  is air density, and the parameter  $C_{DH}$  is the drag coefficient for heat.

2. Page5 Line10: How do you set you simulation time period? Is it the same time period as 1980-2008 of MERRA-2 except the spin up time? Please clarify.

Yes, the time period of the observed results about TPSH and MERRA-2 is from the year 1980 to 2008. As noted in the third comment, observations show a significant decreasing trend form 1970 and an increase trend from 1995 to 2003 in dust storm activities. In order to remove the decadal trend in climate and dust activities, we conducted the CAM4-BAM model with fixed present-day climatological mean SST and sea-ice concentration, as well as greenhouse gases during the 30 simulated years. The simulated year does not represent real time year. Hence, we can only check the relationship between TPSH and dust concentration, and do not compare the simulated results with year by year observations. The corresponding description has been added in the Section 2 "The numerical experiment was conducted with fixed present-day climatological mean sea-ice concentrations, and sea surface temperature (SST), as well as fixed present-day greenhouse gases during the whole simulated period." and "A numerical experiment was integrated over 37 years with 7 years for spin up, including the aerosol direct effect and snow-darkening effect of absorbing aerosols. Note that the simulated year does not represent real time year, hence we can only check the relationship between TPSH and dust concentration, and do not compare the simulated results with year by year observations."

3. Section 3.2: Researches have shown that the dust emission trend in East Asia is decreasing since 1970 but with a small increase trend from 1995 to 2003. The results in Figure 3f is consistently showed this small increase trend while the model results in Figure 3c didn't. This is also reasonable that the correlation coefficient in Figure3F is much smaller than that in Figure 3c. In this case, the authors should investigate and

try to explain the difference with the information of other factors such as the activity of the polar vortex, an important factor that influence the cold air mass activities then hence the dust emission activities in the targeting region.

Yes, I agree with the Reviewer's comment. Observations show a significant decreasing trend form 1970 and an increase trend from 1995 to 2003 in dust storm activities. Hence, there exists a decadal change in dust storm activities, which is mainly related to polar vortex activities (Qian et al., 2002; Zhao et al., 2004; An et al., 2018). Our observed result also show similar decadal change in dust concentration, especially in Figure 3f, as mentioned by the Reviewer. In order to remove the decadal trend in climate and dust, we conducted the CAM4-BAM model with fixed present-day climatological mean SST and sea-ice concentration, as well as fixed greenhouse gases during the 30 simulated years. The simulated year does not represent real time year. Hence, we can only check the relationship between TPSH and dust concentration, and do not compare the simulated results with year by year observations. Hence, our simulated results do not have the decadal change in dust activities, compared with observed results in Figure 3f.

4. Section 3.3: In figure 6d-6f, there are remarkable dust emissions in west and central TP, this not consistent with the normal dust sources distributions in East Asia in other researches. Maybe there are some errors in the source data, please clarify.

Yes, it is evident that there are no large deserts over the western and central TP. However, there exist many aeolian desertified lands over the region (Li et al., 2018). These aeolian desertified lands over the western and central TP can contribute regional dust emissions. Our simulated result is similar with the recent results from the Chinese Unified Atmospheric Chemistry Environment for Dust (CUACE/Dust) (in Figure 2, An et al., 2018), which is an operational mesoscale numerical model to forecast sand and dust storms in East Asia.

5. Section 4: All the meteorological results related to the TPSH's potential mechanisms of the effects on dust cycle are deduced from the model result. Have you

ever evaluated the basic parameters of temperature and wind with the routine meteorological measurements or the reanalysis in the targeting region? As the differences in temperature and in wind from Figure7-10 are quite small which may be in the same range of or be noised by the bias of the modelling.

Yes, the updated CAM4-BAM model has been evaluated against CRU, MODIS data or NCEP2 reanalysis for surface temperature, snow cover, and atmospheric circulation in our recent works (Xie et al., 2018; Shi et al. 2019). These results shown that the model can mainly capture the spatial pattern of these meteorological variables including surface temperature, snow cover and atmospheric circulation. Secondly, our results about differences between strongest and weakest TPSH in atmospheric temperature and atmospheric circulation are absolutely consistent with ones based sensitivity experiment with and without TPSH (Figures 7 and 8 in Duan et al., 2017). Based on these two points, we believe the feedbacks of temperature and atmospheric circulations due to TPSH are reliable.

6. Page9 Line28-29: As there is no detail evidence from the manuscript for the dust deposition on SNOW and so for the feedbacks, please be careful to draw the conclusion.

I have added one figure about dust-in-snow effect and the corresponding discussions in the Section 5. Our results show that, compared to the weakest TPSH years, the MAM dust cycle in the strongest TPSH years are much more vigorous over East Asia. In the strongest TPSH years, much more dusts deposited on snow over TP (Figure 12a) show larger dust-in-snow forcing (Figure 12b) and then further enhance the regional dust cycle through the above positive feedback loop of dust-in-snow.

Figure S2. Spatial distribution of the MAM composite difference between the strongest and weakest TPSH years (strongest-weakest) for the model in (a) dust mass in snow column (g m-2) and (b) dust-in-snow forcing (W m-2). The green-contour area indicates the plateau above 2500 m.

1. Page3 Line 34: Is the 'CAM-BAM' right? Should it be 'CAM4-BAM'? Please check.

Taken.

2. Page5 Line 27-28: For the sentence ' It presents ..., 30 model years, respectively', it is repeated described, please delete it.Taken.

3. Page8 Line18-20: For the sentence 'According to ... with the significant increase in TPSH', it is not logically right. Please find another way to describe it and make it more readable.

Taken

Page 19: The 'C" is missing in the figure caption.
Taken.

5. Page 21: Please use the same dust concentration scale for Figure 3b and Figure 3e,

for Figure 3c and figure 3f, to make it more readable. Do you mean the dots in the black box? They are not very clear in Figure 3a and Figure 3d as the dots look like some lines. Please clarify.

Yes, we have revised the Figure 3 with the same scale for dust concentrations for Figure 3b and Figure 3e, and for Figure 3c and figure 3f, respectively. Additionally, the dots should be the slanted lines and have been revised.

---

## Author Comment (AC2) · 4 Aug 2020

Manuscript Number: acp-2019-393

Journal: ACP

The revised manuscript entitled "Modulation of springtime surface sensible heating over the Tibetan Plateau on the interannual variability of East Asian dust cycle" by Xiaoning Xie, Anmin Duan, Zhengguo Shi, Xinzhou Li, Hui Sun, Xiaodong Liu, Xugeng Cheng, Tianliang Zhao, Huizheng Che, and Yangang Liu

We thank the ACP Handing Editor (**Professor Kari Lehtinen**) for his hard work and the two anonymous referees for their constructive suggestions to improve our manuscript significantly. We greatly appreciate the generally positive comments from both the two Reviewers (Reviewer #1 and Reviewer #2), and have addressed all the concerns, with point-by-point responses detailed below (reviewers comments in red color and our responses in blue color). We have uploaded the file of "Response to reviewers.pdf".

Best wishes,

Xiaoning Xie

Response to Reviewer #1:

General comments:

Dust has important effects on TP and Asian regional climate, while the change in TP climate can feed back to dust generation and lifecycle. The authors conducted model simulations combined with analysis of observations and reanalysis datasets to quantify the effects of TPSH on the interannual variability of dust life cycle in East Asia. This would potentially enhance our understanding of the interactions between dust aerosols and Asian/TP climate, and hence regional climate change. The manuscript is generally well-structured, and the methodology is also sound. Before it can be considered for publication, I have a few comments and suggestions to potentially improve the quality of the manuscript. Particularly, more clarifications and discussions are needed in the modeling and analysis parts. Please see my specific

comments below.

Response: Thank the Reviewer #1 very much for the positive comments. According to the comments, we have added some descriptions in the modeling and analysis parts, and some discussions in Section 5 in responses to the specific comment below.

1, Title: The authors mainly focused on MAM TPSH, so I suggest including "Springtime" in the title to avoid any confusion.

Taken. According to the Reviewer's comment, we have changed the title as **"Modulation of springtime surface sensible heating over the Tibetan Plateau on the interannual variability of East Asian dust cycle"**

2, Introduction: It would be good if the authors could explicitly highlight the difference and novelty of this study compared with their previous study (Xie et al., 2018b), since there are some overlaps between the two studies.

Yes, I agree with the Reviewer' comment. Based on the sensitivity of the GCMs, the enhanced surface sensible heat flux can enhance the Asian dust cycle through the dust-in-snow radiative forcing in Xie et al., 2018b. Hence, we want to know, in the interannual variability of observations and models, whether the springtime surface sensible heating over the Tibetan Plateau can affect the variability of East Asian dust cycle to further confirm this mechanism. Hence, in our manuscript, we have added "More recently, based on sensitivity of GCM simulations, Xie et al. (2018b) revealed that the dust-in-snow radiative forcing over TP significantly increases the eastern Asian dust emissions and the regional dust cycle through enhancing the TPSH, indicating a positive feedback loop. However, the detailed relationships between TPSH and the East Asian dust cycle remain elusive, especially for interannual variability. Therefore, we should check the relationship with springtime TPSH and Asian dust cycle Asian dust cycle to further confirm this mechanism, based on the interannual variability of observations and models." in Section Introduction.

3, Section 2: (1) The website links for the observation and modeling data used in this study need to be provided if they are publicly accessible. (2) For the model, I suggest the authors use CAMchem/MAM in the future study, which includes more realistic aerosol representations. Besides, the assumption of aerosol external mixing and/or simplified (i.e., spherical) particle structure/morphology in CAM4-BAM can lead to uncertainty/bias in DRF (e.g., He et al., 2015: https://doi.org/10.5194/acp-15-11967-2015; Scarnato et al., 2015: https://doi.org/10.5194/acp-15-6913-2015). This issue needs to be discussed to some extent in the manuscript. (3) Was the aerosol-cloud interaction included in the model? The authors did not mention this. (4) I am not quite convinced that it is a good idea to fix the emissions in year 1850. As the authors mentioned, the anthropogenic emissions could serve as a confounding factor for the perturbation of the MAM TPSH impact. If removing the anthropogenic emissions, then the resulting effect of TPSH on dust cycle would not be realistic. If so, the authors need to make some clarifications in the title, abstract, and introduction sections to state that this study only considers the scenario without anthropogenic emission effect. This issue is particularly important when considering that the observations used to evaluate model actually reflect both the natural dust emissions and anthropogenic emissions of other species throughout the study period.

Yes, I believe that the reviewer's comments are very constructive for improving our manuscript. Firstly, all the website links for data of CAM4-BAM, MERRA-2 reanalysis, and the surface sensible heating flux of 73 meteorological stations of the China Meteorological Administration over TP used in this study have been shown in **Section Data availability**. We have added the corresponding descriptions as "The surface sensible heating flux of 73 meteorological stations of the China Meteorological Administration over TP is available in the personal hompage of Anmin Duan, at http://staff.lasg.ac.cn/amduan/index/article/index/arid/11.html. The MERRA-2 reanalysis are developed by the GMAO with support from the NASA Modeling, Analysis and Prediction program, acquired from https://goldsmr5.gesdisc.eosdis.nasa.gov/data/ (last access: 14 Sept. 2018). Simulated

data of CAM4-BAM can be made accessible on request to the corresponding author (xnxie@ieecas.cn).”

Secondly, according to the Reviewer's comments about (2) and (3), we have added the description in Section 5 "The CAM4-BAM model assumed a subbin fixed size distribution of externally mixed aerosols and spherical particle structure/morphology of dusts (Neale et al., 2010). This simplified assumption of aerosols lead to uncertainty or bias in evaluating dust direct radiative forcing (Yang et al., 2007; He et al., 2015; Scarnato et al., 2015). In microphysical processes of the model, the cloud droplet number concentration and ice number concentration are fixed as constant, ignoring aerosol-cloud interactions (Neale et al., 2010). The model cannot evaluate dust effects on warm, mixed or ice phase clouds. Additionally, the assumed dust-snow external mixing underestimates the dust-in-snow feedbacks in CAM4-BAM, and the new parameterization in dust-snow internal mixing enhances the radiative feedbacks (He et al., 2019). Hence, exact parameterizations with dust optical properties, dust-cloud processes and dust-in-snow interactions will reduce the model uncertainty and can effectively evaluate dust-climate interactions in the future."

Finally, we agree with the Reviewer's comments about effects of anthropogenic aerosols. The anthropogenic aerosols (especially black carbon aerosols) can deposit on TP, which affects the TP surface albedo and the surface sensible heat flux (e.g., Qian et al., 2011; Shi et al., 2019). Additionally, according to the Reviewer's comments, we also conducted the experiment with the anthropogenic aerosol and precursor gas emissions at the year of 2000 (PD). Figure S1 shows the spatial distribution of the correlation coefficients between the TPSH and the anomalies of surface dust concentration in the new experiment with PD aerosol emissions. It is shown that the spatial pattern is very similar with Figure 3 in our manuscript. Therefore, the relationship between TPSH and surface dust concentration in interannual variability is exactly true, although the anthropogenic aerosols can affect the TPSH. Hence, we believe that the experiment with fixed PI aerosols is suitable to investigate the effect of springtime TPSH on the interannual variability of East Asian

dust cycle.

[Figure]

Figure S1. Spatial distribution of the correlation coefficients between the index of sensible heat over the TP (TPSH index) and the anomalies of surface dust concentration in MAM for the 30 year CAM4-BAM simulation. Here the slanted lines in the grey (a, d) represent the grid points where the changes pass the two-tailed t test at the 5% significance level and the green-contour area indicates the plateau above 2500 m.

4. Section 3.1: (1) Some more discussions on the reasons of the spatial patterns and biases of TPSH need to be provided. Currently, the authors mainly described the results without many explanations of the physics behind in this section. (2) A particularly interesting question is that dust also feeds back to the TP climate and affect TPSH. For example, column dust leads to surface dimming and reduce TPSH, while dust in snow enhances surface warming and increase TPSH. So how does this dust feedback contribute to the variation of TPSH under the effects of other influential factors? The authors mainly discussed the effect of TPSH on dust in the following sections, but it will be interesting to see the effect of dust on TPSH too. (3) The authors seem to focus on evaluating modeled surface dust concentrations only. Why not also evaluate the column property such as AOD? This at least will be meaningful

over dust source regions. This is also important for the discussions on dust loading in the following sections.

Yes, I agree with the Reviewer' comments and have added the corresponding descriptions. Firstly, we have added the corresponding explanations about the spatial distribution and interannual variations of TPSH in the manuscript. We have been added "The spatial patterns of the MAM TPSH are basically consistent with the ground measurements, reanalysis, and satellite data (Shi and Liang, 2014). There exist persistent snow cover over the western TP and several mountains (Pu et al., 2007; Xie et al., 2018b), which increases surface albedo and modulates the radiative energy balance, and then leads to lower sensible heat fluxes (Xie et al., 2005; Wang et al., 2014)." and "The result indicates a significant interannual variation of the MAM TPSH, which are mainly account for that of surface wind speed, ground-air temperature, and snow cover/depth in winter-spring (Duan et al., 2011; Shi and Liang, 2014; Wang et al., 2014)." Secondly, the point about dust effect on TPSH from the Reviewer is very good. I have added the corresponding discussions about dust direct and dust-in-snow effect (Figure S2 in the followings) on TPSH in the Section 5. "Over the TP, the dust-in-snow forcing is dominated compared with dust direct forcing, which determines the TP warming and regional dust cycle (Qian et al., 2011; Xie et al., 2008b). A significant feature of dust-in-snow effect over the TP creates a positive feedback loop enhancing the East Asian dust cycle. Our results show that, compared to the weakest TPSH years, the MAM dust cycle in the strongest TPSH years are much more vigorous over East Asia. In the strongest TPSH years, much more dusts deposited on snow over TP (Figure 12a) show larger dust-in-snow forcing (Figure 12b) and then further enhance the regional dust cycle through the above positive feedback loop of dust-in-snow." Finally, the modeled dust AOD of CAM4-BAM has been evaluated against Measurements from global in Figure 4 (Albani et al., 2014) and eastern Asian region in Figure 3 (Xie et al., 2018a). Hence, the corresponding description has been added in the manuscript "For the dust aerosol optical depth (AOD), the CAM4-BAM has been evaluated against ground-based measurements from global scale (Albani et al., 2014) and eastern Asian region (Xie et

al., 2018a), showing strong and positive correlations with observational sites on seasonal and annual means."

[Figure]

Figure S2. Spatial distribution of the MAM composite difference between the strongest and weakest TPSH years (strongest-weakest) for the model in (a) dust mass in snow column (g m$^{-2}$) and (b) dust-in-snow forcing (W m$^{-2}$). The green-contour area indicates the plateau above 2500 m.

5. Section 5: (1) The authors made the argument that the dust-in-snow feedback plays an important role, but without any quantitative analysis and/or figures. The dust-in-snow feedback is actually an interesting point, so it would be good to see some quantitative results, figures, and discussions to back up the argument. Besides, a recent study (He et al., 2019: https://doi.org/10.1029/2019MS001737) showed that the dust-snow albedo effect/feedback can be significantly enhanced by dust-snow internal mixing compared with external mixing (presumably assumed in CAM4-BAM model). This could potentially enhance the importance of the dust-in-snow feedback in modulating TPSH. Some discussions on this aspect would be useful. (2) Since a number of potential uncertainty factors (some of them have been mentioned in my comments above) could be involved in the model simulations and analysis, I suggest including a paragraph or two to specifically discuss the uncertainties of this study in this section.

Yes, the constructive suggestions have been provided from the Reviewer about Section 5. Firstly, we have added Figure 12 (Figure S2 in the followings) about

dust-in-snow forcing and the corresponding discussions about dust-snow external mixing in CAM4-BAM. Secondly, according to the Reviewer's comments, we have added a paragraph about potential uncertainty factors of the CAM4-BAM in Section 5 as "The CAM4-BAM assumed a subbin fixed size distribution of externally mixed aerosols and spherical particle structure/morphology of dusts (Neale et al., 2010). This simplified assumption of aerosols lead to uncertainty or bias in evaluating dust direct radiative forcing (Yang et al., 2007; He et al., 2015; Scarnato et al., 2015). In microphysical processes of the model, the cloud droplet number concentration and ice number concentration are fixed as constant, which ignoring aerosol-cloud interactions (Neale et al., 2010). The model cannot evaluate dust effects on warm, mixed or ice phase clouds. Additionally, the assumed dust-snow external mixing underestimates the dust-in-snow feedbacks in CAM4-BAM, and the new parameterization in dust-snow internal mixing enhances the radiative feedbacks (He et al., 2019). Hence, exact parameterizations with dust optical properties, dust-cloud process and dust-in-snow interactions will reduce the model uncertainty and effectively evaluate dust-climate interactions in the future."

References

Albani, S., Mahowald, N. M., Perry, A. T., Scanza, R. A., Zender, C. S., Heavens, N. G., Maggi, V., Kok, J. F., and Otto-Bliesner, B. L.: Improved dust representation in the Community Atmosphere Model, J. Adv. Model. Earth Sy., 6, 541–570, https://doi.org/10.1002/2013MS000279, 2014.

He, C., Liou, K.-N., Takano, Y., Zhang, R., Levy Zamora, M., Yang, P., Li, Q., and Leung, L. R.: Variation of the radiative properties during black carbon aging: theoretical and experimental intercomparison, Atmos. Chem. Phys., 15, 11967–11980, https://doi.org/10.5194/acp-15-11967-2015, 2015.

He, C., Liou, K.-N., Takano, Y., Chen, F., and Barlage, M.: Enhanced snow absorption and albedo reduction by dust-snow internal mixing: modeling and parameterization, J. Adv. Model. Earth Sy., 11, 3755-3776, https://doi.org/10.1029/2019MS001737, 2019.

Qian, Y., Flanner, M., Leung, L., and Wang, W.: Sensitivity studies on the impacts of Tibetan Plateau snowpack pollution on the Asian hydrological cycle and monsoon climate, Atmos. Chem. Phys., 11(5), 1929–1948, https://doi.org/10.5194/acp-11-1929-2011, 2011.

Scarnato, B. V., China, S., Nielsen, K., and Mazzoleni, C.: Perturbations of the optical properties of mineral dust particles by mixing with black carbon: a numerical simulation study, Atmos. Chem. Phys., 15, 6913-6928, https://doi.org/10.5194/acp-15-6913-2015, 2015.

Shi, Z., Xie, X., Li, X., Yang, L., Xie, X., Lei, J., Sha, Y., and Liu, X.: Snow-darkening versus direct radiative effects of mineral dust aerosol on the Indian summer monsoon onset: role of temperature change over dust sources, Atmos. Chem. Phys., 19, 1605–1622, https://doi.org/10.5194/acp-19-1605-2019, 2019.

Wang, Z. Q., Duan, A. M., and Wu, G. X.: Time-lagged impact of spring sensible heat over the Tibetan Plateau on the summer rainfall anomaly in East China: case studies using the WRF model, Clim. Dynam., 42, 2885-2898, 2014.

Xie, L., Yan, T., Pietrafesa, L. J., Karl, T., and Xu, X.: Relationship between western North Pacific typhoon activity and Tibetan Plateau winter and spring snow cover, Geophys. Res. Lett., 32, L16703, doi:10.1029/2005GL023237, 2005.

Xie, X. N., Liu, X. D., Che, H. Z., Xie, X. X., Wang, H. L., Li, J. D., Shi, Z. G., and Liu,Y.: Modeling East Asian dust and its radiative feedbacks in CAM4-BAM, J. Geophys. Res. Atmos., 123, 1079-1096, https://doi.org/10.1002/2017JD027343, 2018a.